# Machine and Deep Learning Approaches Applied to Classify Gougerot–Sjögren Syndrome and Jointly Segment Salivary Glands

**DOI:** 10.3390/bioengineering10111283

**Published:** 2023-11-03

**Authors:** Aurélien Olivier, Clément Hoffmann, Sandrine Jousse-Joulin, Ali Mansour, Luc Bressollette, Benoit Clement

**Affiliations:** 1ENSTA Bretagne, Lab-STICC UMR CNRS 6285, 29200 Brest, France; aurelien.olivier@ensta-bretagne.org (A.O.);; 2GETBO UMR 13-04 CHRU Cavale Blanche, 29200 Brest, France; 3CROSSING IRL CNRS 2010, Adelaide 5005, Australia

**Keywords:** machine learning, deep learning, texture analysis, radiomics, classification, multi-supervision, ultrasound imaging, Gougerot–Sjögren syndrome

## Abstract

To diagnose Gougerot–Sjögren syndrome (GSS), ultrasound imaging (US) is a promising tool for helping physicians and experts. Our project focuses on the automatic detection of the presence of GSS using US. Ultrasound imaging suffers from a weak signal-to-noise ratio. Therefore, any classification or segmentation task based on these images becomes a difficult challenge. To address these two tasks, we evaluate different approaches: a classification using a machine learning method along with feature extraction based on a set of measurements following the radiomics guidance and a deep-learning-based classification. We propose, therefore, an innovative method to enhance the training of a deep neural network with a two phases: multiple supervision using joint classification and a segmentation implemented as pretraining. We highlight the fact that our learning methods provide segmentation results similar to those performed by human experts. We obtain proficient segmentation results for salivary glands and promising detection results for Gougerot–Sjögren syndrome; we observe maximal accuracy with the model trained in two phases. Our experimental results corroborate the fact that deep learning and radiomics combined with ultrasound imaging can be a promising tool for the above-mentioned problems.

## 1. Introduction

This study addressed the problem of the clinical diagnosis of Gougerot–Sjögren syndrome (GSS) using ultrasound imaging. The Gougerot-Sjögren syndrome is an autoimmune disease that involves an inflammatory process and lymphoproliferation that primarily affects the lacrimal and salivary glands. Common symptoms include xerostomia (dry mouth), keratoconjunctivitis sicca (dry eyes), and enlargement of the parotid gland [1]. This disease affects about 0.5 to 4. 8% of the population [2] and specifically females; indeed, more than 90% of the patients affected by GSS are women [2,3].

Primary GSS (pGSS) concerns patients who have only been affected by GSS; if they suffer from another autoimmune disease, then it is called secondary GSS. Unfortunately, actual medical treatment can only relieve the symptoms of the disorder. The challenges in the diagnosis of SSG are related to the fact that its symptoms can be confused with systemic diseases, such as sarcoidosis, amyloidosis, IgG4-related disease, HIV, and lymphoma. In fact, these systemic diseases affect the salivary and lacrimal glands and can cause syndromes similar to GSS [4,5].

The salivary glands facilitate the production of saliva, mastication, swallowing, speech, and taste perception. There are three main salivary glands and 600 to 1000 small minor salivary glands that can be present throughout the mouth [6]. Three main salivary glands exist on both sides of the face: the sublingual gland is the smallest and closest to the mouth, the submandibular gland, and the parotid gland, located near the ear, is the largest.

Usually, GSS detection is performed via biopsy. However, Cornec et al. [7,8] reported a strong correlation between biopsy results and salivary gland ultrasonography (SGUS). Their study motivated several researchers to further investigate non-invasive diagnosis approaches using SGUS. Additionally, deep learning approaches consist of promising tools to tackle challenging tasks in image analysis.

Our project aimed to improve the automatic detection and segmentation of GSS in SGUS using machine learning and deep learning methods. Recent studies show that deep learning approaches may produce strong performance and are robust to environmental changes. However, the results obtained using deep learning methods may, at certain levels, be explainable [9,10], with a challenging hyperparameter tuning that requires a large dataset. Moreover, radiomics have recently been applied to many medical image analysis tasks, and they showed understandable and explainable features. Therefore, we developed a machine learning method optimized via radiomic guidance [11].

## 2. Related Works

In this section, existing GSS detection methods are briefly discussed, and then we present several models applied to texture analysis and deep learning methods for image analysis. This section also presents the deep learning approaches dedicated to biomedical image analysis.

Several studies have been published on the segmentation and classification of GSS from ultrasound imaging. Two approaches have been used for automatic classification, either training a deep neural network [12], or using feature extraction methods combined with a machine learning classifier [13,14]. To characterize GSS and extract some useful features, Berthomier et al. [13] proposed an approach based on a scattering operator. Other works used features based on grey-level textures and statistics, following radiomics approaches [11,15,16]. Deep learning methods have produced exciting results in several fields of medical ultrasound imaging, such as in the evaluation of neuromuscular disease [17].

To segment glands using ultrasound imaging with deep neural networks, Vukicevic et al. [18] compared various architectures of deep convolutionnal neural networks (CNNs), such as U-net and FCDenseNet, for salivary gland segmentation in a database of 1184 ultrasound images obtained from 287 patients, all of who were diagnosed with primary GSS. In their study, they achieved a Dice score (The Dice score is used to gauge model performance, ranging from 0 to 1. It can characterize the number of true positives; at the same time, it penalizes the algorithm for its false positives) of 0.91 with FCN8.

To detect infected glands using ultrasound images, Kise et al. use the VGG16 network [12] pretrained on ImageNet [19]. They classified GSS into four classes with the approval of some experts: definitely GSS, probable GSS, probablt not GSSS, definitely not GSS. The authors used a database containing 200 acquisitions to train their model and obtained an area under curve (AUC) (The ROC (receiver operating characteristic) curve is a graph showing the performance of a classification model at all classification thresholds. The AUC corresponds to the area under the ROC, which indicates the true positive rate against the false positive rate for various classification thresholds) of 0.810 for the parotid glands and an AUC of 0.894 for the submandibular glands [12].

Table 1 and Table 2 present several texture analysis methods from a historical point of view and a performance analysis [20] over 13 texture datasets. We mention the best results obtained on the Textures under varying Illumination, Pose and Scale (KTH-TIPS) image database and KTHTIPS2b in Table 2.

The concept of texture was introduced and questioned by Julesz in 1962 [21]. The author proposed a decomposition of textures into basic spatially local primitives called textons. Texture can be characterized by the occurrence of a pattern along the spatial distribution, taking into account the intensities of the pixels [27], which makes it useful for the classification of similar regions in different images. The specificity of a spatial pattern is that it cannot be defined on a single pixel compared to color. Texture analysis was adopted for the analysis of echocardiograms in 1983 [28]. In 1973, Haralick et al. [22] extracted statistical features on the co-occurence of pixel intensities in an image. In 1989, Mallat introduced the wavelet transform (A wavelet transform approximates an image by dilated and translated local wavelets) to analyze texture in both the frequency and spatial domains. In 2002, Ojala et al. proposed a descriptor called the local binary pattern that uses the sign of the difference between a pixel and the neighboring pixels. Let *P* be the number of neighboring pixels, *R* is the radius that defines the neighborhood, *s* is the sign operator, and $q_{c}$ is a pixel in the image. The descriptor $LBP_{P,R}$ is computed as follows for every pixel $q_{c}$:(1)$$LBP_{P,R}\left(q_{c}\right)=\sum_{p=0}^{P-1}s\left(g_{p}-g_{c}\right)2^{p}$$

The histogram of the descriptor can then be used to identify patterns. The limitation of descriptors is caused by the sign operation, which loses local information. In 2012, Bruna et al. [25] proposed scattering convolutionnal networks, invariant to translation and rotation, using a cascade of wavelet transform convolutions with modulus and low-pass filters. Cimpoi et al. developed a Fisher vector-convolutionnal neural network (FV-CNN) to remove global spatial information. Song et al. [29] improved the model by adding local connected layers and a loss layer using the hinge loss (The hinge loss is used for “maximum-margin” classification, most notably for support vector machines). This model produced state-of-the-art results on the KTH-TIPS2 data set [30].


bioengineering-10-01283-t002_Table 2Table 2Texture analysis methods that obtained an accuracy >99% on KTHTIPS challenge [20].MethodSorted random projections [31]Scale-invariant feature transform [32]Scattering convolutionnal networksVGG [33]Locally transferred Fisher vector for classification (LFVCNN) [29]


Since 2012, deep learning has become popular in image processing, with the first human-competitive results achieved on a digit classification dataset [34,35,36]. Performance was then improved using deeper networks, requiring convolutionnal layer reformulation, such as residual learning [37], densely connected neural networks [38], dilated networks [39], or using attention mechanisms [40,41]. In 2021, Dai et al. [42] achieved the best performance on ImageNET classification benchmarks with a hybrid model that combines attention and convolutionnal networks.

Deep learning methods have been extensively applied to medical image analysis in segmentation or classification tasks [43,44]. Ronneberger [45] developed the U-net model using convolutionnal blocks with several resolutions, downsampling and upsampling paths, and skip connection mapping layers at the same resolution from both paths. This method and recent improvements with attention blocks have provided strong performances on small datasets [45,46,47].

## 3. Databases

This section describes the databases used in our project. In addition, we describe the preprocessing steps introduced for each database. We also introduce the new salivary gland database of SGUS gathered in our project, called GSID, which contains 210 SGUS cases of normal and ill submandibular glands, provided by the Brest University Hospital Center (Brest UHC). We also used an open source database, named Harmonicss, of SGUS [48]. Using the Harmonicss database, we used two test methods to evaluate the adaptation of our models to a new environment. The first method directly predicts the model trained on the GSID. In the second method, we created training, validation, and testing sets from the Harmonicss database, fine-tuning our model on the training set, and reproducing it with a 10-fold cross-validation.

The GSID database contains patients with suspected GSS. Several images were annotated by experts with dot points and others with a binary mask segmentation. We developed the following algorithm to create a segmentation mask from a dot–points contour. The images in the database are either in grey scale with a white dot–point contour or in red, green, blue (RGB) images with a color dot–point contour. We transformed the RGB images in hue saturation values (HSV) and applied a threshold on the color of the dot points using the following operations (Figure 1):HSV transformationColor thresholdGaussian blurDilationFilling holesErosion

For the grey-scale images, we applied a threshold on the white color (Figure 2) and used the following operations:White thresholdDetecting isolated componentsRemoving large isolated componentsDilationFilling holesErosionBlur

A threshold is a simple operation that selects grey-level values between 200 and 255, while the Gaussian blur applies a Gaussian filter of size (3, 3). The dilation and erosion operations are morphological operations based on a structural element given by a binary matrix. The dilated (respectively, eroded) value at a given pixel *x* is the maximum (respectively, the minimum) value in a window, defined by the structuring element, centered on *x* [49]. The images are finally resized to a size of (192, 192) via bilinear interpolation.

The open-source database of the Harmonicss project (HARMONIzation and integrative analysis of regional, national, and international cohorts on primary Sjögren’s syndrome (pSS) toward improved stratification, treatment and health policy making) contains 225 ultrasound images of 225 patients from 4 European centers [50], see Table 3. The salivary glands in the acquisition are parotid or submandibular collected by four different devices (Samsung, Philips, esaote, GE), see Figure 3. The database includes for each patient:DeVita’s score [51]: The DeVita score ranges from zero to three in each gland, from normal-appearing morphology to severe inhomogeneity.The Outcome Measures in Rheumatology Clinical Trials (OMERACT) score [52]: The OMERACT score proposes a four-grade classification.The classification of the European League against Rheumatism (EULAR) [53]: The EULAR score is based on focal lymphocytic sialadenitis with a focus score (FS) ≥1 by LSG, presence of anti-SSA(Ro) antibodies, positive ocular staining score (OSS ≥5), positive Schirmer’s test (≤5 mm/5 min), and unstimulated whole salivary (UWS) flow rate ≤ 0.1 mL/min.The duration of the disease.

To compare the scores given in this database with the binary classification used for the GSID database, we applied a threshold on the scores. We call devita0 the binary score created from the DeVita score, in which we assigne the score in the range [1, 3] of the DeVita score to label 1 and a score of 0 to label 0. Similarly, we call devita1 the score created by assigning the range [0, 1] in the DeVita score to 0 and the range [2, 3] to 1.

## 4. Methods

In this section, we present different detection approaches tested on SGUS. We used two different classes of methods applied to GSS detection:The first method computes textural features following the radiomics guidance and trains a classifier based on these features.The second method uses a deep neural network with an innovative 2-phase training scheme with pre-training based on joint classification and segmentation loss.

Radiomics feature extraction first requires the extraction of a region of interest (ROI) in which to compute the features. Three cases occur (see Figure 4): Firstly, ROI annotations are present and a deep neural network can perform strong automatic annotations. Secondly, partial annotations are present on the images, but the segmentation task is too hard to reproduce; finally, there is no annotation given and no segmentation network.

While we made segmentation models for GSS, our final approach for assessing the potential of radiomics feature extraction consisted of annotations to generate an ROI mask. In this assessment, we only considered the data where experts’ annotations were provided. We also trained a deep neural network to automatically provide the segmentation mask with or without annotations.

### 4.1. Features for Texture Description

This part of our work focused on combining the extraction of texture features with machine learning classifiers. We selected first-order statistical features and other features extracted from 4 matrix-containing texture information: grey-level co-occurrence matrix (GLCM) [22], grey-level run length matrix (GLRLM) [54,55], grey-level dependence matrix (GLDM) [56], and grey-level size zone matrix (GlZSZM) [57]. Statistical characteristics were computed from these 4 matrices (see Table 4). We define the GLCM and GLRLM matrices and a part of those features in the next section.

#### 4.1.1. Grey-Level Quantization

The texture features presented in the previous section depend on the quantization of the grey-level distribution. We computed the features using a discretization, with specific criteria for the selection of the histogram bins. The first method possible for the bin selection, called bincount, is based on a fixed number of bins to create this histogram, whereas the second method, called bin width, uses a fixed width for every bin in the histogram. Following the documentation of PyRadiomics, we used the bin width method, testing grey-scale-range width values of 10, 25, and 50, aiming to reach a bincount of 30 to 130 bins. Lofstedt et al. [58] recently modified Haralick features to be invariant to grey-level quantization. We describe below how the statistical features are extracted from the texture matrix constructed on the discretized grey levels.

#### 4.1.2. Grey-Level Co-Occurrence Matrix

The main idea of this method is based on the relationship between texture and tone in grey-scale-level analysis. An image can be divided into zones, where either tone or texture can be dominant. The method focuses on the size of these zones and their contained elements, such as the number of distinct tones, the element length, or the size of the zone. We assumed that texture information can be described by the matrix that counts the co-occurrence of grey-scale levels, with a fixed distance and angle. Let *M* be a matrix containing four grey levels (1–4):$$M=\left[\begin{array}{cccc} 2 & 1 & 2 & 3\\ 1 & 1 & 1 & 3\\ 3 & 4 & 3 & 2\\ 2 & 4 & 4 & 4 \end{array}\right]\;$$

The co-occurrence matrix is defined as follows, where N(0,1) is the number of co-occurrences of tones 0 and 1 for a given angle and distance:$$A=\left[\begin{array}{cccc} N(1,1) & N(1,2) & N(1,3) & N(1,4)\\ N(2,1) & N(2,2) & N(2,3) & N(2,4)\\ N(3,1) & N(3,2) & N(3,3) & N(3,4)\\ N(4,1) & N(4,2) & N(4,3) & N(4,4) \end{array}\right]\;$$

The GLCM characteristics were calculated on the matrix of the four following angles (0°, 45°, 90°, 135°) and averaged. We show the formulas of 5 of the 22 GLCM features calculated (see Table 4). We define a(i,j) as the coefficients of the normalized matrix *A* and $N_{g}$ as the number of discrete intensity levels in the image. $\sigma_{x}$ is the standard deviation of the marginal row probability. Hereinafter, we define several features:Marginal row probability $p\left(i\right)=\sum_{j=1}^{N_{g}}a\left(i,j\right)$Joint average: $\mu_{x}=\sum_{i=1}^{N_{g}}\sum_{j=1}^{N_{g}}a\left(i,j\right)i$Angular second moment: $f1=\sum_{i=1}^{N_{g}}\sum_{j=1}^{N_{g}}a{\left(i,j\right)}^{2}.$Contrast: $f2=\sum_{i=1}^{N_{g}}\sum_{j=1}^{N_{g}}a\left(i,j\right){\left(i-j\right)}^{2}.$Correlation: $f3=\frac{\sum_{i=1}^{N_{g}}\sum_{j=1}^{N_{g}}ija\left(i,j\right)-2\mu_{x}}{2\sigma_{x}}.$Sum of squares: $f4=\sum_{i=2}^{2N_{g}}\sum_{j=1}^{N_{g}}{\left(i-\mu_{x}\right)}^{2}a\left(i,j\right).$Entropy: $f5=-\sum_{i=1}^{N_{g}}\sum_{j=1}^{N_{g}}a\left(i,j\right)\log\left(a\left(i,j\right)\right).$

#### 4.1.3. Grey-Level Run-Length Matrix

Whereas the GLCM matrix focuses on the number of grey-level co-occurrences, then Glrlm matrix counts each grey level, i.e., the length of segments of consecutive pixels of a single grey- evel. Let *M* be a matrix of four grey levels (1–4) as defined earlier. A grey-level run is a set of consecutive, collinear points with the same grey-level value. For a given picture, we computed a grey-level run-length matrix *A* for runs having any given direction. A matrix element a(i,j) specifies the number of times that the picture contains a run of length *j* in the given direction, consisting of points having grey-level *i* [59]. From this matrix, we reproduced the features following the radiomics guidance. We chose to present three of all descriptors obtained from this Glrlm matrix:Short-run emphasis
(2)$$\frac{\sum_{i=1}^{N_{g}}\sum_{j=1}^{N_{r}}\frac{a(i,j)}{j^{2}}}{\sum_{i=1}^{N_{g}}\sum_{j=1}^{N_{r}}a\left(i,j\right)}$$Long-run emphasis
(3)$$\frac{\sum_{i=1}^{N_{g}}\sum_{j=1}^{N_{r}}j^{2}a\left(i,j\right)}{\sum_{i=1}^{N_{g}}\sum_{j=1}^{N_{r}}a\left(i,j\right)}$$Run percentage
(4)$$\sum_{i=1}^{N_{g}}\sum_{j=1}^{N_{r}}a\left(i,j\right)$$

To enhance the classifier performance, we normalized all the features using the following:Standard: Letting μ be the mean and σ be the standard deviation for a feature *x*, we computed the standard normalization:
(5)$$\frac{x-\mu }{\sigma }$$Min–max: We computed the Min–max normalization for any feature *x* relative to the max and min of all feature values.
(6)$$\frac{x-min(x)}{max(x)-min(x)}$$Principal component analysis (PCA): For PCA normalization, we selected the number of components such that the variance was greater than 0.95 [60].

### 4.2. Classifier and Feature Selection

In this section, we present the feature selection method and the classifiers used in order to obtain the best accuracy (*ACC*) (Equation (Equation 7)), where *TP* = True Positives, *TN* = True Negatives, *FP* = False Positives, and *FN* = False Negatives:(7)$$ACC=\frac{TP+TN}{TP+FP+TN+FN}.$$

To select the most relevant classifier among support vector machine [61], random forest, and regression, we performed a 10-fold cross-validation for each model.

#### 4.2.1. Random Forest Feature Selection

A random forest consists of the randomized training of an ensemble of trees. A tree is defined as a structure containing nodes in which the input data are split into two subsets according to a threshold on a variable of the input data. In order to measure the importance of a variable, we used the mean decrease impurity importance (MDI) [62,63]. MDI averages, over all trees and for all splits using one variable, the difference between the Gini impurity (Gini impurity measures the frequency for a randomly selected and labeled element if a set is incorrectly labeled, according to the distribution of labels in the set) of a node and the sum of the Gini impurity of its child nodes weighted by the probability that a sample may reach each node.

#### 4.2.2. Maximum Relevance, Minimum Redundancy

Maximum relevance, minimum redundancy (MRMR) consists of selecting the features with the highest relevance for the classification, while minimizing the redundancy of the selected features [64,65]. Letting $N_{s}$ be the number of features in the subset *S*, the relevance was computed as follows:(8)$$D=\frac{1}{N_{s}}\sum_{x_{i}\in S}I(x_{i};c)$$
where I(x,y) stands for the mutual information:(9)$$I\left(x;y\right)=\int \int p\left(x,y\right)\log\frac{p(x,y)}{p(x)p(y)}dxdy.$$

Then, redundancy is computed as follows:(10)$$\;\;R=\frac{1}{N_{s}^{2}}\sum_{x_{i},x_{j}\in S}I(x_{i},x_{j}).$$

To maximize the relevance and minimize the redundancy, we maximized the following function:(11)maxΦ(D,R)=D−R.

To find the most relevant features for classification, this method was applied to the database. We chose the same number of features as obtained with the random forest feature selection.

### 4.3. Deep Neural Network with Joint Training Scheme

In the previous section, we presented the method used to train a machine learning classifier with features extracted on a texture matrix with statistical descriptors. This section presents the second method used for the classification of GSS. We describe the used deep neural networks, the training scheme, and the selected hyper-parameters.

#### 4.3.1. Model

Using five layers of double convolutionnal blocks with 4 downsampling applied within the 5 layers and a reconstruction path for the segmentation part with upsampling operations, our baseline model is inspired by U-net [45]. That model is based on double convolutionnal blocks and skip connections mapping the downsampling and upsampling paths.

A double convolutionnal block consists of blocks formed by a convolutionnal layer with a batch normalization operation and a ReLU regularization [66]. To expand the receptive field, we used dilated convolutions in the second layer of each double convolutionnal block, without losing resolution or increasing the number of parameters [39]. This method allows end-to-end training of a deep neural network for classification with a relatively small database.

Internal covariate shift is defined by the changes in the distribution of network activation due to changes in network parameters during training. It can produce very low or very high values, which can lead to vanishing or exploding gradients. To avoid this distribution shift issue, we used the batch normalization method. This normalization method can accelerate the training of deep neural networks and improve performance [67,68]. For each batch, the mean and variance are computed to normalize the data. Then, the scaling parameter γ and the shifting parameter β are learned to produce the batch normalization:(12)$$y_{i}=\gamma {\hat{x}}_{i}+\beta$$

This function is applied on every feature map at each batch to train the parameters and perform batch normalization.

#### 4.3.2. Joint Training Scheme

This section describes the multi-phase joint-training scheme used to train the deep neural network. We considered several training schemes. Joint training is a case multi-task learning [69], which has been largely considered in deep learning but rarely in medical imaging. In our case, joint-training is seen as a way to increase performance, facilitate optimization, and improve generalization, under the assumption that the segmentation task and classification task require similar features. However many multi-task problems are considered only as a requirement for the objective task. In multi-task learning optimization, Yu et al. [70] identified three concepts, conflicting gradients, dominating gradients, and high curvature. Conflicting gradients occur when gradients from multiple tasks are in conflict with one another; dominating gradients occur when the difference in gradient magnitudes is large, leading to some task gradients dominating others; and high curvature occurs in the multi-task optimization landscape. We assumed that in our case, gradients have less risk to consider, but there is a risk of dominating gradients between the segmentation loss and the classification loss. To deconflict gradients during optimization Yu et al. [70] proposed a procedure called PCGrad, in which the gradient of each task is projected onto the normal plane of the gradient of the other tasks.

Inspired by the Y-net proposed by Mehta et al. [71], we built a joint segmentation and classification network. The novelty of our proposal, called two-phase training, is the use of this joint training as pre-training in a first phase, then a second phase trains the model specifically on the classification loss in a classical way. The joint-training phase uses the classification branch along with the segmentation branch to regularize the model during 200 epochs. Then, the model is specifically trained on the required task, either using the classification branch alone or the segmentation branch alone for 600 epochs. This regularization is used to push the network to generate high-level task-relevant features using the low-level features built in the first phase, to thus fine tune the network more specifically. We did not freeze any layer during the second training phase, and the final classification model had the same number of parameters as the first-hase network.

During the first phase, called the joint training scheme, the classification branch is linked to the lowest-resolution layer at the bottleneck of the segmentation model, see Figure 5. The classification part contains a global pooling operation that produces a vector, whose size is the number of filters at the bottleneck layer. Cross-entropy is used as classification loss, and a weighted sum of a cross-entropy function and a Dice score produces the segmentation loss. During the second phase of 600 epochs, we only used the class categorical cross-entropy to train the network. We describe the loss functions used in more detail in the next section.

#### 4.3.3. Training Settings

Our loss function is based on the sum of cross-entropy and the Dice score. Let $x_{i}$ and $y_{i}$ be, respectively, the binary predicted label and the ground truth label for a pixel i∈[1,n], where *n* stands for the number of pixels. Cross-entropy is defined as:(13)$$l_{ce}=-\sum_{i=1}^{n}\left(y_{i}log\left(x_{i}\right)+\left(1-y_{i}\right)log\left(1-x_{i}\right)\right)$$

Cross-entropy was firstly introduced as a cost function to train deep neural networks for the segmentation of biomedical images [45]. However, this metric is not robust to unbalanced data when a class is under-represented [72]. On the other hand, the Dice score provides improved performance when applied on an unbalanced dataset for several tasks [73,74,75]. Following the scheme proposed in [46], we used a weighted sum of the Dice score $L_{segDice}$ and the cross-entropy $L_{segCE}$ to compute the segmentation loss. We also used the cross-entropy $L_{classifCE}$ to compute the classification loss:(14)$$L_{total}=coef\;f_{ce}*L_{classi\;f}+coef\;f_{{seg}_{dice}}*L_{segDice}+coeff_{{seg}_{ce}}*L_{segCE}$$

For other training hyperparameters of the deep neural network, as described in Table 5, we used 32 filters in the first layer and 600 epochs of training. To learn the weights of the network, the optimizer for the stochastic gradient descent was based on the Adas method, which adds momentum to the classical algorithm [76]. The learning rate, which weighs the optimization step, was set to $10^{-5}$.

Data augmentation is commonly used to tackle the lack of data by applying various image transformations. To ensure the robustness of the implementation, we used the package imgaug. The transformations were consistently applied to labeled image. The used transformations were left and right flip, cropping, sharpening, affine transformation, linear contrast, Gaussian blur, additive Gaussian noise, edge detection, dropout, and elastic transformation [77]. The augmentations were performed sequentially during the training, and we randomly selected three augmentations to be applied for each batch.

### 4.4. Metrics

To measure and interpret the classification results, we computed the accuracy, sensitivity, and specificity as follows:(15)$$\begin{array}{ccc} Sensitivity & = & \frac{TP}{TP+FN} \end{array}$$(16)$$\begin{array}{ccc} Specificity & = & \frac{TN}{TN+FP} \end{array}$$(17)$$\begin{array}{ccc} Accuracy & = & \frac{TP+TN}{TP+TN+FP+FN} \end{array}$$

To summarize, this section presented the method inspired by radiomics guidance, as well as an innovative training scheme for deep neural networks using joint training on classification and segmentation, the initialization of the model weights for 200 epochs in order to train the model to learn the segmentation task features, and a specific training phase on classification. Next, we compare this method to classical training on the classification task and compare the results with those produced by machine learning models trained with texture and statistical descriptors.

## 5. Simulation Results

For classification and segmentation tasks, we present, in separate sections, the results of the two above–mentioned methods: the extraction of radiomics features combined with a machine learning classifier and deep neural networks. For GSS detection with deep learning, we provide the results obtained on experiments with various hyperparameter sets on our labeled database and then the results obtained on an unseen unlabeled database gathered by the Harmonics project.

### 5.1. Classification Results with Radiomics Features

We present here the results obtained for the selection of the best pixel normalization and bin width, the features selected with these parameters using 10-fold cross-validation, and the accuracy metric. We compare as well feature selection methods with various feature sets that measure the mean of all classifier accuracies and the maximal accuracy obtained across all classifiers.

By performing the bin-width selection with a random forest classifier and PCA feature normalization, we obtained a best accuracy of 0.79 with a sensitivity of 0.839 and a specificity of 0.776 with a bin width of three (see Table 6). We observed a slight decrease in the sensitivity with a bin width of 5 and a decrease in specificity with a bin width of 10. The bin width of 25 provided the lowest performance. These results showed that a more precise discretization produced more relevant features, good overall performance using the classifier and radiomics features with a relevant discretization, and good balance between sensibility and specificity. Taking into account these results, we fixed the bin width to five for the following experiments to compare classifiers and normalization methods.

The comparison of the 10-fold classification results over all features with a fixed bin width, for various classifiers and normalizations, is given in Table 7. This table shows that the best performance was obtained with random forest and standard normalization considering the accuracy metric, with 0.789 compared to 0.782. The SVM classifier with standard normalization provided a good specificity of 0.816 but a slightly lower accuracy, with 0.771 against 0.789 with random forest.

We tested two feature selection methods with various numbers of features. As 23 features were selected using MDI, we fixed the same feature number for the MRMR feature selection to compare the methods (see Table 8). The best 10-fold accuracy of 0.84 was obtained with STD image normalization and logistic regression without a feature normalization.

The tests of various feature selections on the GSS detection showed better results for all classifiers with the features selected by random forest (MDI), as well as the maximum accuracy obtained with one classifier. The MRMR-selected parameters provided a slightly lower mean accuracy over all classifier and a decrease of 0.04 for the best classifier compared with that of the model using features selected with MDI. Both feature selections provided an improvement in the mean and max accuracy compared with the classifiers trained using all features. Additionally, using the 10 most important features according to MRMR provided better results than using the 23 most important features.

In this section, we compared the detection results of GSS with various classifiers, data normalization, and bin widths. This work allowed us to find the best settings for this method and to obtain an accuracy that was next used as a basis for a comparison with the results obtained from a deep neural network.

### 5.2. Classification Results with Deep Neural Network on GSS Detection

In this section, we present the results of various experiments conducted using deep neural networks using the best, precise hyperparameters described in the previous sections. We also compare the two-phase training results to classical training results in terms of classification loss. Based on our database, we compared our results to the ones of a machine learning classifier. Additionally, we present the results of a model trained on our database and used to detect GSS on Harmonicss.

We performed a twofold cross-validation with two launches for each set of hyperparameters, as indicated in Table 9.

Table 10 shows that the two-phase model had slightly higher accuracy (0.019) on two-fold cross-validation averaged over input image shapes of (192, 192) and (128, 128).

According to Table 11, the best accuracy of 0.905 was obtained with a coeff_classif_ce of 1, a coeff_seg_dice of 0.5, and a coeff_seg_ce set to 0. Similar accuracy was obtained by increasing coeff_seg_dice to 1. A lower accuracy was obtained for all coeff_seg_dice values lower than 0.5.

The best results were obtained with neighboring values for the classification cross-entropy and the segmentation Dice. However, lower values were obtained when adding the segmentation cross-entropy loss or when using a small coefficient for the segmentation Dice loss.

Over all hyperparameter sets and using the two best loss coefficient combinations, as defined in Table 11, the accuracy differed slightly, with an accuracy 0.008 higher for the two-phase training (see Table 12). The results obtained during this experiment are shown in Figure 6. The three best and the two lowest precisions appeared with the two-phase model.

The maximum accuracy was obtained without normalization with a coefficient of 1 for the classification loss and a coefficient of 0.5 for the Dice segmentation loss (See Table 13).

The results obtained with the best hyperparameters for the two-phase training of the deep convolutionnal neural network were higher by 0.16, with a 1.0 accuracy for the two-phase-trained DCNN, compared with 0.84 for the best classifier trained on radiomics features (See Table 14).

By applying the model trained in the GSID database to the Harmonicss database, the best accuracy obtained was 0.831 using the devita1 score and an image preprocessed to a shape of (192, 192) (See Table 15). The accuracy obtained on the devita0 score was 0.08 lower, at 0.751. The specificity obtained using the devita0 score was slightly higher than when using devita1, with 0.918 for devita0 versus 0.899 for devita1. However, the sensitivity was much higher for devita1 with 0.724 versus 0.591 for devita0.

The results obtained on the original Harmonicss images without preprocessing were very low, with an accuracy of 0.547 obtained with devita0. A preprocessing method called ‘Adapted’, which reshapes images adapted to all different input shapes, achieved a lower accuracy than reshaping all images to (192, 192). The accuracy decreased by 0.08 from 0.921 on the GSID test set compared to 0.831 on the Harmonicss database.

Table 16 shows the results obtained after fine tuning the model on the Harmonicss database. We used five different training and testing splits in order to obtain a five-fold cross-validation. In the fine-tuning case, the testing set was a split of the Harmonicss database, and the results were averaged over five different splits of the database. In the direct prediction case, the testing set was the entire Harmonics database.

Using the trained model, direct predictions on the Harmonicss database compared with the devita0 score showed an accuracy of 0.756, with a low sensibility and a high specificity. Using the devita0 classification score, a fine tuning of the same model achieved an accuracy of 0.873, which was 0.117 higher than that of direct predictions, with a much higher sensibility of 0.900 compared with 0.667. The direct predictions on Harmonicss compared with the devita1 score produced an accuracy of 0.831. The fine tuning of the same model performed using devita1 classification produced a high accuracy of 0.929, which was 0.092 higher than that achieved with direct predictions. This showed that on both metrics, fine tuning consequently increased the accuracy. Additionally, the accuracy on the devita1 score was higher by 0.075 than the devita0 score for direct prediction, whereas it was higher by 0.056 for the predictions after fine tuning.

## 6. Discussion

This paper presents an analysis of several GSS detection methods using ultrasound imaging. The experiments aimed to find the best classifier, normalization, bin width, and most relevant features for the methods based on radiomics. We also compared two deep neural network training schemes with US as the input. This included a hyperparameter search and a search for the training loss coefficient. This allowed us to compare the deep learning method to a classical machine learning method and perform a robustness test of the model on an unseen database.

The classification tests using radiomics features showed that the feature selection improved the accuracy up to 0.18 on average and up to 0.4 at a maximum. The feature selection based on random forest classifiers generated the best average and maximal accuracies over all tests. This highlights the importance of feature selection in the training phase of machine learning classifiers on radiomics features. The feature selection method has a large impact on accuracy. Additionally, an optimal bin width was found for the grey-level quantization on this task. These results were used to compare the machine learning methods with deep learning methods.

The best parameters for the two-phase training were obtained with an input image shape of (128, 128) and without any image normalization. Concerning the loss coefficient, adding the cross-entropy loss to the segmentation penalization did not result in any improvement. Additionally, the best accuracies were obtained with a classification coefficient of 1, a segmentation cross-entropy coefficient of 0, and a Dice coefficient of 0.5 or 1. Therefore, a Dice loss coefficient close to the classification loss coefficient produced better results. In general, the best accuracy was obtained with a dice loss coefficient of 0.5 with an accuracy of 1.

Two-phase training and “class” training were tested on the same hyperparameter sets with two-fold cross-validation. The mean accuracy over all tests differed only by a slight increase of 8‰ for the two-phase model. The best global results were surprisingly obtained without any image normalization with a perfect accuracy on one-fold cross-validation. On the hyperparameter sets with no normalization, the accuracy obtained with the joint two-phase model was 1.8% higher than that of the one-phase model trained on classification. Furthermore, the jointly trained two-phase model was only 1‰ higher than the jointly trained one-phase model. A potential improvement was achieved using the two-phase model on images without normalization. However, there was no significant improvement between the models trained with joint loss on one or two phases. This can be explained by all layers being retrained in the two-phase model and that the model can efficiently learn relevant features to both classification and segmentation.

Based on accuracy, the best results were obtained in the salivary glands database, with the two-phase model achieving a two-fold cross-validated accuracy of 1.0 compared with the classification based on radiomics features with 0.84. These results show that both methods are relevant for the detection of GSS. However, differences between the two methods could be further observed on unseen datasets. The deep neural network trained on GSID adapted well to an unseen environment on the open-source part of the multicentric Harmonicss database, with an accuracy of 0.831 based on the devita1 score. Furthermore, better results were obtained for devita0 and devita1 after a fine tuning of the model originally trained on a GSID training set and retrained on the Harmonicss dataset, using devita0 or devita1 scores as labels. The increase in accuracy obtained with the fine tuning was larger on devita0 than on devita1. Therefore, the model can quickly adapt to the desired classification score.

Our performed experiments compared models based on machine learning to deep learning and tuned the best parameters. Whereas both radiomics and deep learning features seemed relevant for our application, we found that the deep learning models are more accurate due to the larger possibility of encoding shifts in images and environment. The second key of our approach is the joint segmentation pretraining in the two-phase model, which did not increase performance over all hyperparameter sets but increased the maximal accuracy. Hence, the learning of relevant segmentation features produced relevant classification features that were not learned by the single-phase model and helped reach the maximal accuracy. A deeper study of the model performance on an unseen database would help to conclude if the gain obtained is specific to the used database.

## 7. Conclusions

Based on joint classification and segmentation training, this study developed the first two-phase joint training of a deep convolutionnal neural network and compared it to two other methods: classical training with classification loss and a machine learning classifier using textures features as the input. Classification using radiomics features showed good performance for GSS detection. But the deep neural network trained for classification was more accurate than the machine learning model based on radiomics features. The innovative two-phase training scheme showed the best improvement compared with the single-phase model without any normalization and provided a perfect accuracy on the test set. The results obtained were globally similar for the previous approach and a model trained with classification only. However, our new method can be useful when both segmentation and classification are required. It would be worth testing this training scheme on a task where the classification model produces a lower accuracy around 0.7 to observe the higher contribution of the two-phase or multi-phase training.

Finally, the single phase deep neural network performed well on the unseen Harmonicss dataset obtained from an open-source multicentric study, which showed that the model is robust to new domains. This opens good perspectives for the application of this algorithm in clinical settings. Future work will consider data fusion between the radiomics coefficient and the features of deep neural networks.

## Figures and Tables

**Figure 1 bioengineering-10-01283-f001:**
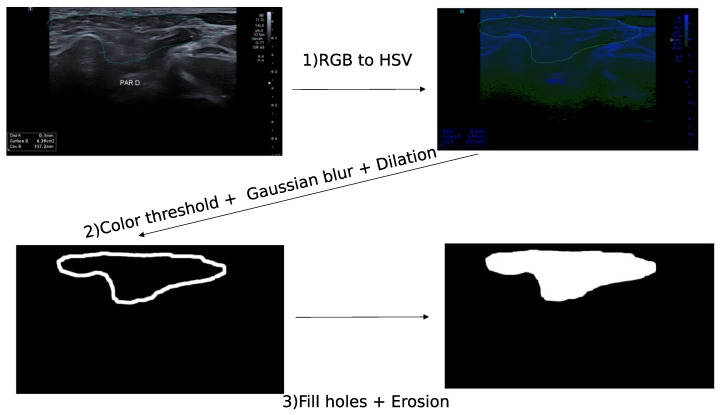
Annotation of contour on the RGB imgaes from GSID database. The first image represents the ultrasound image encoded in RGB values, the second image is the RGB image transformed in HSV, and the third image is the contour obtained with a dilation a Gaussian blur and a color threshold. The image on the bottom left is the annotated segmentation after filling the contour and an erosion.

**Figure 2 bioengineering-10-01283-f002:**
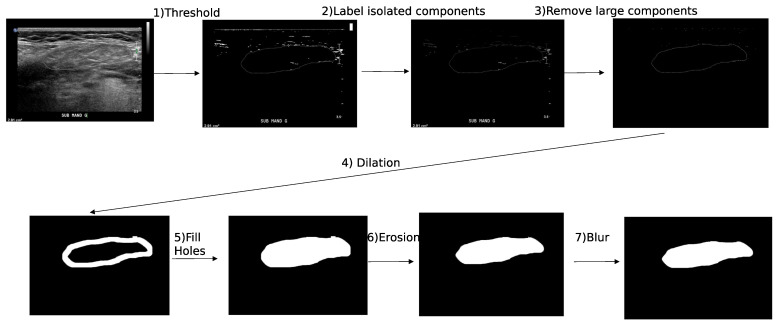
Annotation of the contour on grey-scale annotation.

**Figure 3 bioengineering-10-01283-f003:**
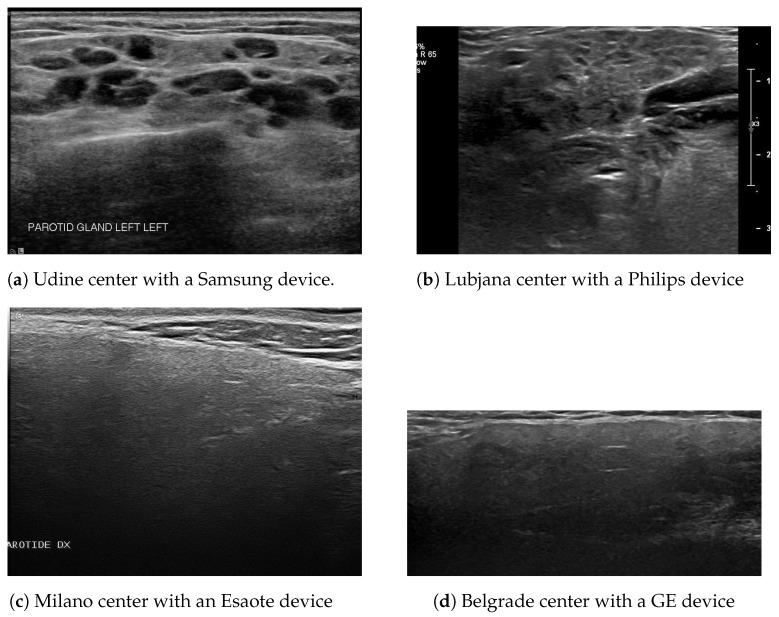
Examples of images getting from different centers and using various devices.

**Figure 4 bioengineering-10-01283-f004:**
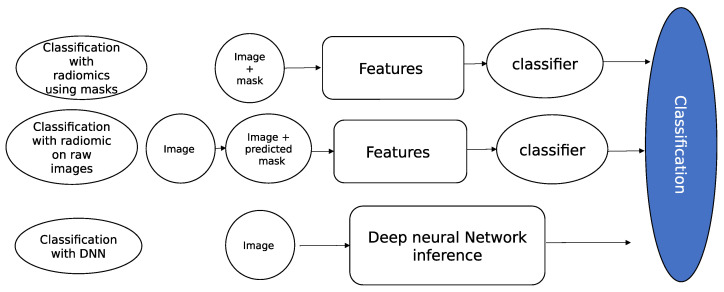
Three different classification schemes and the steps in each scheme: classification with radiomics features using masks, classification with radiomics features with DNN-predicted mask, and classification with a DNN.

**Figure 5 bioengineering-10-01283-f005:**
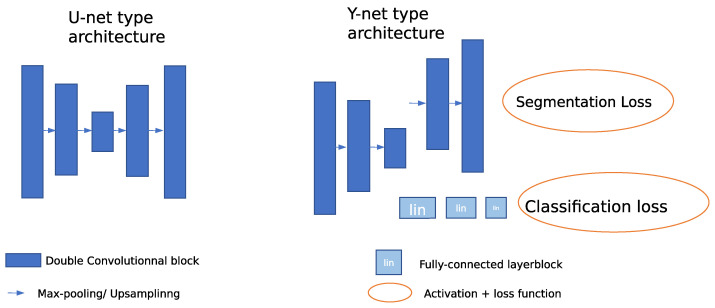
Coarse representation of the architectures of U-net [45] and Y-net with 2 downsampling layers. The skip connections are not included in the figure for simplification. The decrease in the height of the convolutionnal block denotes upsampling or downsampling.

**Figure 6 bioengineering-10-01283-f006:**
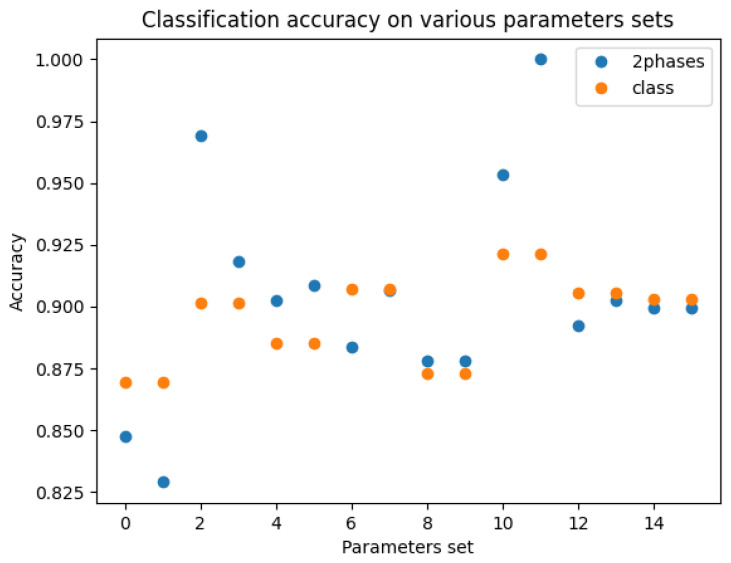
Classification accuracy for all sets of hyperparameters: blue, 2-phase model; orange, “class” with the classification only model.

**Table 1 bioengineering-10-01283-t001:** Overview of the evolution of texture analysis methods.

Method	Year	Author
Texture perception model [21]	1962	Julesz
Gray-level co-occurence matrix (GLCM) [22]	1973	Haralick et al.
Wavelet [23]	1989	Mallat
Local binary pattern for facial texture [24]	2002	Ojala et al.
Scattering convolutionnal networks [25]	2012	Bruna et al.
Fisher vector pooling of CNN [26]	2015	Cimpoi et al.

**Table 3 bioengineering-10-01283-t003:** Description of the Harmonicss database.

Database Entry	Range Value or Value
Patients number	225
Gland type	parotid/submandibular
Ultrasound	4 constructors
DeVita	[0, 3]
OMERACT	[0, 3]
Disease duration	[0, 24 weeks]
EULAR	[0, 2]

**Table 4 bioengineering-10-01283-t004:** Summary of the features extracted.

Class of Features	Name of Features
GLCM [22]	“Autocorrelation”, “JointAverage”, “ClusterProminence”, “ClusterShade”, “ClusterTendency”, “Contrast”, “Correlation”, “DifferenceAverage”, “DifferenceEntropy”, “DifferenceVariance”, “JointEnergy”, “JointEntropy”, “Imc1”, “Imc2”, “Idm”, “Idmn”, “Id”, “Idn”, “InverseVariance”, “MaximumProbability”, “SumEntropy”, “SumSquares”
GLRLM [54]	“GrayLevelNonUniformity”, “GrayLevelNonUniformityNormalized”, “GrayLevelVariance”, “HighGrayLevelRunEmphasis”, “LongRunEmphasis”, “LongRunHighGrayLevelEmphasis”, “LongRunLowGrayLevelEmphasis”, “LowGrayLevelRunEmphasis”, “RunEntropy”, “RunLengthNonUniformity”, “RunLengthNonUniformityNormalized”, “RunPercentage”, “RunVariance”, “ShortRunEmphasis”, “ShortRunHighGrayLevelEmphasis”, “ShortRunLowGrayLevelEmphasis”
GLDM [56]	“DependenceEntropy”, “DependenceNonUniformity”, “DependenceNonUniformityNormalized”, “DependenceVariance”, “GrayLevelNonUniformity”, “GrayLevelVariance”, “HighGrayLevelEmphasis”, “LargeDependenceEmphasis”, “LargeDependenceHighGrayLevelEmphasis”, “LargeDependenceLowGrayLevelEmphasis”, “LowGrayLevelEmphasis”, “SmallDependenceEmphasis”, “SmallDependenceHighGrayLevelEmphasis”, “SmallDependenceLowGrayLevelEmphasis”
GLSZM [57]	“GrayLevelNonUniformity”, “GrayLevelNonUniformityNormalized”, “GrayLevelVariance”, “HighGrayLevelZoneEmphasis”, “LargeAreaEmphasis”, “LargeAreaHighGrayLevelEmphasis”, “LargeAreaLowGrayLevelEmphasis”, “LowGrayLevelZoneEmphasis”, “SizeZoneNonUniformity”, “SizeZoneNonUniformityNormalized”, “SmallAreaEmphasis”, “SmallAreaHighGrayLevelEmphasis”, “SmallAreaLowGrayLevelEmphasis”, “ZoneEntropy”, “ZonePercentage”, “ZoneVariance”
First order	“10Percentile”, “90Percentile”, “Energy”, “Entropy”, “InterquartileRange”, “Kurtosis”, “Maximum”, “MeanAbsoluteDeviation”, “Mean”, “Median”, “Minimum”, “Range”, “RobustMeanAbsoluteDeviation”, “RootMeanSquared”, “Skewness”, “TotalEnergy”, “Uniformity”, “Variance”

**Table 5 bioengineering-10-01283-t005:** Parameters tested.

Hyperparameter	Searched	Values
Filters	Fixed	32
Augmentations	Fixed	3
Batch size	Fixed	4
Learning rate	Fixed	$10^{-5}$
Training scheme	Searched	[2-phase, class]
Loss coefficients	Searched	[ [10, 0.2, 0.1], [10, 0.2, 0], [1, 1, 0], [10, 0.1, 0], [1, 0.5, 0], [10, 0.01, 0], [10, 0.3, 0] ]
normalization	Searched	[‘no’, ‘standard’]
Image shape (Width, Height)	Searched	[(128, 128), (192, 192)]

**Table 6 bioengineering-10-01283-t006:** Results for various bin widths with a random forest classifier and PCA.

Accuracy	Sensitivity	Specificity	Bin Width
0.79	0.839	0.776	3
0.782	0.823	0.776	5
0.764	0.839	0.731	10
0.738	0.803	0.706	25

**Table 7 bioengineering-10-01283-t007:** Classifier and normalization comparison for a fixed bin width.

Classifier	Feature Normalization	Accuracy	Sensitivity	Specificity
Random forest (RF)	PCA	0.782	0.823	0.776
Random forest (RF)	std	0.797	0.85	0.76
SVM	std	0.771	0.737	0.816

**Table 8 bioengineering-10-01283-t008:** Accuracy (Acc) results obtained with all classifiers with fixed normalization and various feature selection methods.

Feature Selection	Image Norm	Dim	Acc Mean	Acc Max	Acc Std
All features	STD	128	0.705	0.797	0.08
10 selected with MRMR	STD	128	0.721	0.802	0.082
23 selected with MDI	STD	128	0.723	0.84	0.078
23 selected with MRMR	STD	128	0.713	0.79	0.07

**Table 9 bioengineering-10-01283-t009:** Two-phase loss coefficient selection.

Coeff Classif Ce	Coeff Seg Dice	Coeff Seg Ce
1	1	0
1	0.5	0
1	0.03	0
1	0.02	0.01
1	0.02	0
1	0.01	0
1	0.001	0

**Table 10 bioengineering-10-01283-t010:** Two-fold accuracy for two phases or classification phase only without normalization averaged over various image shapes.

Training Phase	Norm	Accuracy
1-phase joint	no	0.908
2-phases	no	0.909
class	no	0.891

**Table 11 bioengineering-10-01283-t011:** Two-phase loss coefficient selection.

Coeff Classif Ce	Coeff Seg Dice	Coeff Seg Ce	Accuracy
1	1	0	0.903
1	0.5	0	0.905
1	0.03	0	0.897
1	0.02	0.01	0.897
1	0.02	0	0.897
1	0.01	0	0.899
1	0.001	0	0.887

**Table 12 bioengineering-10-01283-t012:** Two-fold accuracy for 2 phase or classification phase with all normalizations averaged over various image shapes.

Training Phase	Norm	Accuracy
2-phases	no,std	0.904
class	no,std	0.896

**Table 13 bioengineering-10-01283-t013:** Two-fold max accuracy for 2 phases or classification phase over all hyperparameter sets.

Training Phase	Accuracy
2-phases	1.0
class	0.977

**Table 14 bioengineering-10-01283-t014:** Results of deep-learning- and radiomics-based model on GSID database.

Database	Model	Image Shape	Accuracy	Sensitivity	Specificity
GSID	2-phase	(192, 192)	1.0	1.0	1.0
GSID	Radiomics	(192, 192)	0.790	0.839	0.776

**Table 15 bioengineering-10-01283-t015:** Results of GSS detection with 2-phase model trained on GSID database training set and predicted directly on Harmonicss database.

Database	Score	Image Shape	Accuracy	Sensitivity	Specificity
Harmonicss	devita0	(192, 192)	0.751	0.591	0.918
Harmonicss	devita0	Original	0.547	0.583	0.509
Harmonicss	devita0	Adapted	0.747	0.565	0.936
Harmonicss	devita0	(192, 144)	0.756	0.667	0.955
Harmonicss	devita1	(192, 192)	0.831	0.724	0.899

**Table 16 bioengineering-10-01283-t016:** Accuracy obtained on the Harmonicss database for devita0 and devita1 scores with or without fine tuning with the model trained on GSID.

Score	Fine Tuning	Accuracy	Sensibility	Specificity
devita0	no	0.756	0.667	0.955
devita1	no	0.831	0.724	0.899
devita0	yes	0.873 (5-fold)	0.900	0.853
devita1	yes	0.929 (5-fold)	0.924	0.931

## Data Availability

Not applicable.

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
