# Peer review of "Machine and Deep Learning Approaches Applied to Classify Gougerot–Sjögren Syndrome and Jointly Segment Salivary Glands"

_bioengineering, 2023, doi:10.3390/bioengineering10111283_

Round 1
Reviewer 1 Report
Comments and Suggestions for Authors
In this manuscript, a two-phase neural network method is proposed and used alone with several classical machine learning methods to investigate the potential of ultrasound images in Gougerot-SJögren syndrome classification tasks. However, the novelty of this work is limited, the relevant experiments are inadequate and the writing as well as organization is not logical enough. Therefore, the recommendation of mine is rejection.
Problems are listed as follows:
1. The joint-training uses classification and segmentation maybe has novelty in the field of Gougerot-SJögren syndrome classification, but it has been widely used in other fields.
2. The section of Method needs to be re-organized. The relationship about the subsections 4.1 and 4.2 is not clear. Details about combination of the feature extraction methods and following machine learning classifiers in subsection 4.2 should be given.
3. In the Experiments, classical methods such as VGG, ResNet, Inception should be compared.
4. Contents of Abstract and Introduction need to correspond to the section Experimental. For example, there is no explain why the proposed method can work in the field of constraints arising from the data and modality.
5. Captions and procedure of all Figures should be further improved.
6. The notations in section of Method need to be further improved, the meaning of the symbols used in the formulations should be given. For example, the symbol
in Eqution.1?
7. Vector diagrams are essential.
Comments on the Quality of English LanguageThe expression can be moderate improved.
Author Response
Dear reviewer,
Thank your for your remarks, please find our answers and corrections below.
1)
We agree with the reviewer and a new reference has been added to the modified version of our manuscript; However, and up to our knowledge a Two-Phase Training as suggested in our work has never been done before.
2)
Thanks for your comment, based on this comment, we modified that section by adding missing links between subsections as 4.1, 4.2
3)
We should highlight that the classic and suggested methods such as VGG don't perform well in our case (i.e. with small database).
4)
We agree with the reviewer, we removed the sentence as it is unclear; However we aimed to underline that in our database contains echo-graphic images obtained over years using various machines and collected by different operators, those images are very noisy and sensitive to the operator and the used probes. In previous studies and using advanced signal processing approaches coupled with High Order Statistic tools, but the deep learning methods provide higher robustness to the change in devices and operators.
5)
As by your suggestion, figure captions have been modified.
6)
In this new version, we better defined our variables.
7) Vector diagrams
We modified most of our figures to be in SVG formats except the ultrasound images.
We hope that the new modified version of our manuscript satisfy better your criteria.
Reviewer 2 Report
Comments and Suggestions for Authors
The reviewed article focuses on the utilization of ultrasound imaging (US) as a promising tool for aiding physicians and experts in diagnosing Gougerot-Sjögren Syndrome (GSS). However, challenges arise when attempting classification or segmentation tasks based on these images. To tackle these challenges while considering data and modality constraints, the study evaluates various approaches, including feature extraction methods based on radiomics guidance and deep learning-based classification.
An innovative technique is proposed, involving a two-phase multiple supervision strategy to enhance the training of a deep neural network. This strategy incorporates joint classification and segmentation for pre-training. Notably, the learning methods yield segmentation outcomes comparable to those accomplished by human experts. The study achieves proficient segmentation results for salivary glands and promising detection outcomes for Gougerot-Sjögren Syndrome, with the highest accuracy observed in the two-phase trained model. These findings underscore the potential of combining deep learning and radiomics with ultrasound imaging as a promising approach to address the aforementioned challenges.
Personally, I find the article to be intriguing and recommend its publication with consideration of the following suggestions:
https://www.sciencedirect.com/science/article/abs/pii/S0010482521004170
https://pubmed.ncbi.nlm.nih.gov/37290256/
Emphasize the advantages of ultrasound imaging in musculoskeletal systems further. The following two articles could be referenced to underscore this point:
Provide information regarding the prevalence of Gougerot-Sjögren Syndrome to contextualize its significance.
In line 81, consider revising "introduce" to the past tense to maintain consistent verb tense usage.
Line 87 does not require the repeated use of the full term "Gougerot-Sjögren Syndrome." It could be streamlined for improved readability.
In Figure 1, consider enlarging the letters beside the ultrasound images to enhance legibility.
Address the presentation of Figure 3, as it appears to be poorly edited and challenging to comprehend. Enhancing the clarity of the visual representation would significantly improve its impact.
Author Response
Dear reviewer,
Thank your for your remarks, please find our answers and corrections below.
1)Provide information regarding the prevalence of Gougerot-Sjögren Syndrome to contextualize its significance
Added more clinical context
2)
As by your suggestion and the previous suggestion of the first reviewer, figure captions have been modified.
3) In line 81, consider revising "introduce" to the past tense to maintain consistent verb tense usage.
l81 corrected
4)Line 87 does not require the repeated use of the full term "Gougerot-Sjögren Syndrome." It could be streamlined for improved readability.
l87 corrected
5)In Figure 1, consider enlarging the letters beside the ultrasound images to enhance legibility.
We modified most of our figures.
6) Address the presentation of Figure 3, as it appears to be poorly edited and challenging to comprehend. Enhancing the clarity of the visual representation would significantly improve its impact.
Added captions within the figure and enlarged images
We hope that the new modified version of our manuscript satisfy better your criteria.
Reviewer 3 Report
Comments and Suggestions for Authors
This manuscript developed a new way to train deep neural networks for segmentation and classification for Gougerot-Sjögren syndrom. Specifically, a 2-phases multiple supervision method was developed for the training. The classification results from the deep neural networks were compared with results from machine learning with radiomics features.
Promising results of the deep neural networks were shown for segmenting the salivary glands and diagnosing Gougerot-Sjögren syndrom. While the manuscript showed interesting results and discussion, the following concerns need to be addressed:
- In the abstract, it says, "To address these two tasks with constraints arising from the data and modality, we evaluate different approaches, a feature extraction methods based on a set of measurements following the radiomics guidance and a deep learning based classification. " It seems the paper mainly compared the classification results(i.e. one task) between the two methods. The machine learning with radiomics features only showed classification capability but not segmentation. The author should revise it in the abstract to make it clear.
- The author only mentioned "... the results obtained from an application of a model trained on our database, used to detect GSS on Harmonicss, an unseen database." The manuscript should clearly describe in the Methods section what the 'training data' and 'testing data' are. If a database was split for the training data set/testing data set, what method was used for this splitting?
- Why did Table 12 miss the values in the 'Accuracy' column?
- In Table 14, 2-phases deep learning was compared with radiomics using the GSID database. What are the results comparing the two using the harmonics database? If the GSID database was only used for training, a comparison of the two methods using the testing data, 'Harmonicss', is of more interest to the audience.
- In the Conclusion section, the author mentions, ''Based on the accuracy, the best results are obtained on the salivary glands database, with the 2-phases model with an 2-fold cross-validated accuracy of 1.0 compared to the classification based on radiomics features with 0.84.". The author should clarify it by saying "the best results on the training data set is..."
- In Table 10,12,13,14, there are '2-phases', '2phases', and '2-phases joint'. Are they the same?
- What is 'joint-net' in Table15 caption. Should it be '2-phases trained DCNN'?
- The authors need to check the grammar of the paper. For example, '...segmentation task is to hard to reproduce...' in Line 143, Page 6; '...aiming to reach a of a bincount from 30 to 130 bins.' in Line 165, Page 6; etc.
- In addition, the authors should check the capitalization rules of words and ensure a consistent capitalization rule is used throughout the texts and the figures.
Author Response
Dear reviewer,
Thank your for your remarks, please find our answers and corrections below.
1)
We should highlight that radiomic features have been considered in the classification part of our approach and not in the segmentation part. To clarify that fact, we enhanced that in the new version of our manuscript.
2)
Thanks for your comment, to clarify our ideas, we modified our text by adding " In our study and using the harmonicss database, two test methods have been implemented and used:
- The first method consists in a direct prediction of the model trained on the GSID training split.
- In the second method, we created a training, validation and testing sets from the harmonicss database. Then, we conducted a fine-tuning on our model using the test set and validated the result with a 10-fold cross-validation. "
3)
Many thanks for mentioning that, we correct these typos.
4)Table 14 2phases deep learning compared with radiomics using the GSID database. What are the results comparing the two using the harmonicss database.
This is an interesting remark, however we don't have the ground truth segmentation for the harmonicss database, thus the extraction of the radiomics features relies on a predicted segmentation. In addition, the radiomics method already produce a lower accuracy on the GSID database, hence we were more interested about the capacity of the deep learning model to be as performant on new datasets.
5)
We clarified that the results are given on the test set.
6)
Yes we changed the notation consistently to : 2-phases
7)
Thank you for your remark, we changed the caption
Round 2
Reviewer 1 Report
Comments and Suggestions for Authors
The recommendation of mine is minor revision. Although most of the raised issues have been revised, there are still problems could be improved:
1. The introduction to the background of Gougerot-Sjögren syndrome seems too cumbersome.
2. The author should thoroughly review and rectify any grammatical and spelling errors in the article.
3. It would be better if Figure.5 could be optimized with more details.
Comments on the Quality of English LanguageThere are some typos should be corrected in the article, and it should be consistent in verb tense.
Author Response
Thank you for your reply,
We added more context about the Gougerot-Sjögren syndrome on the request of another reviewer.
We reviewed all paper for grammatical and spelling errors
We added more details on the figure 5
Reviewer 3 Report
Comments and Suggestions for Authors
The authors have addressed all of my comments.
Author Response
Thank you for your reply